# Nanotechnological Interventions in Agriculture

**DOI:** 10.3390/nano12152667

**Published:** 2022-08-03

**Authors:** Zishan Ahmad, Sabaha Tahseen, Adla Wasi, Irfan Bashir Ganie, Anwar Shahzad, Abolghassem Emamverdian, Muthusamy Ramakrishnan, Yulong Ding

**Affiliations:** 1Co-Innovation Centre for Sustainable Forestry in Southern China, Nanjing Forestry University, Nanjing 210037, China; emamverdiyan@njfu.edu.cn (A.E.); ramky@njfu.edu.cn (M.R.); 2Bamboo Research Institute, Nanjing Forestry University, Nanjing 210037, China; 3Plant Biotechnology Section, Department of Botany, Aligarh Muslim University, Aligarh 202002, India; sabahaazeem@gmail.com (S.T.); adlawasi97@gmail.com (A.W.); irfanbashir301@gmail.com (I.B.G.); ashahzad.bt@amu.ac.in (A.S.)

**Keywords:** nanotechnology, nanoparticles, sustainable agriculture, nanomaterials, abiotic stress

## Abstract

Agriculture is an important sector that plays an important role in providing food to both humans and animals. In addition, this sector plays an important role in the world economy. Changes in climatic conditions and biotic and abiotic stresses cause significant damage to agricultural production around the world. Therefore, the development of sustainable agricultural techniques is becoming increasingly important keeping in view the growing population and its demands. Nanotechnology provides important tools to different industrial sectors, and nowadays, the use of nanotechnology is focused on achieving a sustainable agricultural system. Great attention has been given to the development and optimization of nanomaterials and their application in the agriculture sector to improve plant growth and development, plant health and protection and overall performance in terms of morphological and physiological activities. The present communication provides up-to-date information on nanotechnological interventions in the agriculture sector. The present review deals with nanoparticles, their types and the role of nanotechnology in plant growth, development, pathogen detection and crop protection, its role in the delivery of genetic material, plant growth regulators and agrochemicals and its role in genetic engineering. Moreover, the role of nanotechnology in stress management is also discussed. Our aim in this review is to aid researchers to learn quickly how to use plant nanotechnology for improving agricultural production.

## 1. Introduction

Agriculture is one of the most promising sectors to play in the world economy as it provides food for humans and animals and produces raw materials for various industries. For decades, the demographic data of the world population has been constantly changing, and the number of people is expected to exceed 9.7 billion by 2050 [1]. Moreover, according to the estimation of the Food and Agriculture Organization [2], the agriculture sector needs to produce twice as much food to meet the global demands by 2050. The increasing world population, decreasing cultivable land, high rate of deforestation and changing climatic conditions, especially increasing temperature and CO_2_ levels, stress the need to develop new technologies to enhance the yield and productivity of plants during challenging environmental conditions. The challenges of abiotic stress on plant development and growth are among the emerging ecological impacts of climatic change [3]. Plants are fixed and hence exposed to extreme environmental situations such as drought, heavy metals, light intensities, UV, flood, etc., and these stressors from the environment can cause various stress to a variety of species [4,5]. These stressful conditions induce reactive oxygen species (ROS) which in turn cause degradation of the membrane, increase cell toxicity and retard plant growth. Meanwhile, antioxidant systems through enzymatic and non-enzymatic methods remove ROS and relieve stress produced due to oxidation.

However, in field conditions, distinctive effects on plants frequently take place due to this combination of stresses leading to unanticipated physiological effects [6]. Nowadays, various methodologies have been explored focusing on stress tolerance in plants. An effort has been made to breed crops with stress-tolerant traits in the past few decades by considering two important approaches, viz., conventional and mutation breeding. However, the uncertainty of results and the time-consuming process were the drawbacks of these two processes [7]. Introduction of exogenous genes or changing the expression level to improve stress tolerance and obtaining a genetically modified plant is another method [7]. However, this practice is limited and unacceptable in many countries leading to the limitation of this technique. Priming is another approach to make the plants resistant to environmental stresses. Chemical priming is helpful in the establishment of resistance because it induces the existing defense mechanism of the plant without resorting to genetic changes. Pre-treating or priming plants with natural or synthetic compounds enhanced response under stresses such as heat, salinity, drought, etc., in comparison to unprimed plants [8]. The defense mechanism of plants is enhanced by priming due to its ability to activate and amplify those signals which can control the accretion of ROS, redox signaling and expressions of the genes involved in resisting stresses [9]. Most frequently utilized priming agents are amino acids, polyamines, melatonin fungicides, phytohormones, reactive nitrogen, sulfur and oxygen species, etc. [10,11,12].

Today, the application of nanotechnology and its tools in the chemical priming field improved the effectiveness of the chemicals used for priming and thus reduced the chemical release into the environment [13]. Nanotechnology is a novel and innovative approach to develop and design real-world applications of materials at the nanoscale [14,15]. In the agricultural sector, nanotechnology has a great role in dealing with various issues such as making crop plants more resistant to biotic and abiotic stresses and increasing the productivity of the plants. In addition to this, problems associated with the overuse of fertilizers and pesticides and their harmful impact in relation to the environment could also be tackled with a targeted and proper use of nanotechnology [16,17]. Gradually, the nanotechnological intervention in agricultural sectors is increasing and making this sector an income-generating business [18]. Overall, the use of nanotechnology will enhance food quality and global production in an environmentally friendly way by solving the problem associated with water and soil [19,20].

Furthermore, NPs participate in growth and development and also provide protection to the plants. During stress responses, NPs can also modify and change the expression of the genes that cause biosynthesis and organization of cells, electrons and transport of energy [21]. The diverse physicochemical characteristics of nanoparticles is due to their smaller size. They are known for higher reactivity, biochemical activity and solubility due to a higher surface-to-volume ratio [22]. The target-specific and low-quantity release of NPs make them different from various other elements used in plants [23] In addition, the behavior of NPs depends strongly on their chemical composition, particle size and function. Moreover, NPs play a significant role in protecting the plant against different stressors, accelerating the scavenging of ROS, protecting photosynthetic machinery and reducing osmotic and oxidative stress [24,25,26]. Different types of NPs with their unlimited potential to revolutionize the agricultural sector have been used these days along with their benefits and drawbacks [19,27]. From various studies, it was established that nanoparticles are crucial for crop improvement, but their actual working mechanism and mode of interaction are still at an initial stage [28,29].

Nanomaterials (NMs) have become an elemental part of NPs due to their distinctive characteristics in terms of physical, chemical, mechanical and thermal properties. The NMs can be natural or tailored by using physical, chemical or biological methods [30]. In terms of the recent progress, an emphasis has been placed on the use of nanomaterials (NMs) or engineered nanoparticles (ENPs) in agriculture sectors. Nowadays, engineered NPs are used as nanoherbicides and nanopesticides by considering two important factors: controlling the release of agrochemicals with minimized nutrient loss and enhancing plant morphogenesis by targeting particular cellular organs of plants. These days, nanomaterials are present in different aspects of day-to-day life, such as the environment, agriculture, food and cosmetic industry, waste water treatment, medicine, energy information and communication [31,32,33,34,35]. Various NMs such as single-walled carbon nanotubes (SWCNTs), multi-walled carbon nanotubes (MWCNTs), silver, silicon, zinc, iron and titanium dioxide (TiO_2_) have been observed to improve plant growth and development [19,27,36]. Therefore, NPs are involved in different aspects of agriculture such as making the plant resistant to disease and pests, improving nutrient absorption, acting as a carrier for various vital compounds and improving the effectiveness of fungicides, pesticides, herbicides and the delivery of fertilizers leading to a boosted growth of plants [27,37]. However, the use of NMs for the improvement of the crop and sustainable agriculture is still in its initial phase. Hence, to address the problems in agriculture, the knowledge of NPs and their application in agriculture is crucial for workers. It is also mandatory to build up a fundamental understanding of NPs in relation to agriculture. This section briefly describes the role of nanotechnology in different aspects. Considering the importance of nanotechnology, the aim of this review is to summarize the available information and developments on nanotechnological interventions in agriculture. Furthermore, the current communication relates to the use of tailored NMs for sustainable agriculture with a special emphasis on handling plant stress.

## 2. Nanoparticles: Scientific Aspects

The term “nano” is derived from a Greek word that means “dwarf”, and it signifies 10^−9^ parts of any unit [38]. It can be organic or inorganic molecules having dimensions of less than 100 nm and a wide surface area [39]. Plants react differently in the presence of different NPs depending on the size, shape and chemical and physical properties of the NPs [40]. Figure 1 describes the different types of NPs based on function, morphology, chemical structure and physiochemical properties. Applied NPs in agriculture are categorized into three groups, i.e., organic, inorganic (metal and metal oxide NPs) and combined NPs [27]. In most studies, inorganic NPs are used, i.e., metal 25% and metal oxides 54%, while carbon-based NPs are used in only 10% of studies; ZnO, TiO_2_, CuO and CeO_2_ are the most commonly utilized metal oxide NPs, whereas Ag NPs are most commonly used in the metallic NP group [41]. Various other distinctive forms of NPs are core-shell NPs, polymer-coated magnetite NPs, photochromic polymer NPs, Au NPs, Pd NPs and Ni NPs, while some other NPs are metal oxides, for example, MgO NPs, TiO_2_ NPs, CeO_2_ NPs, ZrO_2_ NPs and ZnO NPs. All these NPs have a specific set of properties and can be synthesized by conventional or unconventional approaches [42]. Generally, two processes (top down and bottom up) are employed for the synthesis of NPs [43,44] (Figure 2). Different lithographic techniques, for example, milling, grinding, etc., are employed to break down bulk material into substances at the nanoscale in top-down approaches. While in the case of bottom-up approaches, atoms self-assemble into new nuclei and eventually grow into particles with a nanoscale that also includes physical and chemical methods. Toxic starting materials, high running costs, toxic contamination and high temperature are required in these methods to obtain the final products [43,44]. Several attempts have also been performed to use biological catalysts, viz., plants, bacteria, fungi and yeast, in order to avoid obstacles found in physical and chemical methods [45].

## 3. Nanotechnological Interventions in Agriculture

### 3.1. Role of Nanotechnology in Germination and Growth of Crop

In the past few years, researchers’ interest has increased in nanomaterials and their impact on agriculture. It appears that NMs are involved in the germination of plants and seeds as well as plant growth [34]. Moreover, some researchers have acknowledged that the size of NPs might enhance the breakdown of organic materials and absorption of inorganic compounds, resulting in an improved photosynthetic rate and generation of oxygen-free radicals during photosynthesis [34]. Zheng et al. stated that spinach seeds that were treated with nano-TiO_2_ produced plants that in comparison to non-treated plants consisted of 73% more dry weight, a 45% percent increase in chlorophyll and a 3-fold increase in photosynthetic rate during the germination period of 30 days. In spinach seeds, the germination rate depends on the size of the nanomaterial; hence, the smaller the size, the higher the germination [46]. Ling and Xing investigated NP phytotoxicity and its effect on canola, radish, rapeseed, ryegrass, corn, cucumber and lettuce germination [47]. In ryegrass and maize, the germination was inhibited by a higher concentration of nano-Zn, which was 2000 mg/L, whereas inhibitory effects on root length were produced on 200 mg/L ZnO in the case of the other plants [47]. The effect of four NPs (CeO_2_, Lanthanum (III) oxide-La_2_O_3_, Gadolinium (III) oxide-Gd_2_O_3_, Ytterbium oxide-Yb_2_O_3_) on radish, rape, lettuce, wheat, tomato, cabbage and cucumber root growth was studied by [48]. They concluded that root growth depends on the type and concentration of NPs as supported by the study of Lin and Xing [47]. Moreover, nano-CeO_2_ did not produce any effect on elongation of the root, while, at the same concentration, the other three above-mentioned NPs affected the various stages of root growth, and most of them were inhibitory [48]. Munir et al. studied the effect of ZnO NPs on the seed priming of wheat [49]. They found that the priming of seeds enhanced the total biomass content, photosynthetic efficiency and various other growth criteria of wheat seedlings. Moreover, it was observed that wheat is capable of accumulating ZnO NPs at a higher concentration in its shoot, roots and grains [49]. Similarly, foliar application of ZnSO_4_ and ZnO NPs on the coffee plant resulted in a change in the fresh and dry weight of roots and leaves in comparison to the non-treated plant [50]. A total of 10 mg L^−1^ of Zn either in the form of ZnSO_4_H_2_O or ZnO NPs was used for foliar application on a one-year-old greenhouse-grown coffee plant. It was found that ZnO NPs positively affect the fresh weight of roots by 37% and leaves by 95% as compared to control, and similarly, a dry weight increase was reported as 28% in roots and 20% in leaves, whereas 85% was reported for the stem [50]. In tomatoes, salt stress can be reduced by applying 250 mg L^−1^ of Cu NPs. The decrease in stress might be due to an increase in the ratio of the Na^+^/K^+^ pump. Moreover, the fruit of treated plants contains more Cu, glutathione, vitamin C and phenols in comparison to non-treated plants [51]. Khan et al. evaluated the influence of Ag NPs on various growth parameters of *Silybum marianum* [52]. They observed that the Ag NPs at a lower concentration of 30 μg/mL were able to enhance germination and growth, while at higher concentrations, they showed an inhibitory effect [52]. Similarly, Mehta et al. studied the effect of Ag NPs on *Brassica juncea, Triticum aestivum* and *Vigna sinensis* at different concentrations of 0, 50 and 75 ppm [53]. Moreover, bacteria-synthesized (*Bacillus cereus*, *Brevundimonas diminuta*, *Serratia marcescens*) Ag NPs improve the germination of seeds, length of shoot and root, number of leaves and fresh weight in *Triticum aestivum* [54]. In the germination of rice seeds, the effect of fluorescein isothiocyanate (FTIC)-labeled silica and photostable cadmium-selenide (CdSe) quantum dots were investigated. It was observed that the FTIC-labeled Si NPs improve germination while quantum dots inhibit the rate of germination [55]. Another study on seed-priming-improved germination in *Citrus lanatus* showed that the seedling emergence rate at 14 days was higher in Ag NP treated seeds as compared to the other treatment [56]. They found that glucose and fructose content was enhanced during germination in Ag NP treated seeds at the interval of 96 h. In a recent study on *Vigna radiata,* it was found that TiO_2_ NPs from a seed extract of *Trachyspermum ammi* significantly improved the growth of *V. radiata* in both in vitro and in vivo conditions [57]. Application of carbon nanomaterials, viz., CNTs and CNPs, stimulated corn seed germination with Cu stress alleviation [58]. They found that both CNTs and CNPs have a positive effect, but the improvement made by CNPs on plant growth was found to be stronger than CNTs. Similarly, seed priming with MWCNTs modulates plant growth and seed germination in two maize varieties under cadmium stress [59]. The study found that MWCNT application increases the germination rate by 11.42% and 24.76% in both varieties, and along with this, an increase in shoot and root fresh weight and antioxidant enzyme activity was also reported. Overall, the studies reported above showed that seed priming with NPs could enhance seed germination growth and rate while maintaining the quality of the plant/fruit using sustainable nanotechnological approaches.

### 3.2. Role of Nanotechnology for Delivery of Plant Growth Regulators (PGRs)

PGRs are classified under a special family of pesticides, are synthesized artificially and play a very crucial role in plant growth and development at very low concentrations. They belong to the five major categories similar to phytohormones, viz., auxin, cytokinins, gibberellins, ethylene and abscisic acid (ABA). Both phytohormones and PGRs are similar in structure and physiological effects but different in their derivations [60]. The combination of NMs and PGRs is employed in agriculture in detecting the trace amount of plant hormones in the plant and its availability to obtain a better response. In addition, it is also found that NMs help in the absorption and transport of PGRs into the plant. One of the characteristic features of sustainable agriculture is the efficient delivery of agrochemicals and organic molecules. Usually, broadcasting or spraying is used to add agrochemicals to the crops. These methods are not appropriate because most of the pesticides, fertilizers and phytohormones are wasted and only a small amount, which is far below the actual required amount, reaches the crops. Hydrolysis, photolysis, chemical leaching and microbial decays are the major factors responsible for the losses. Therefore, in recent years, the use of NPs has increased for agrochemical delivery as they provide efficient transfer systems due to their wide surface area, ease of binding and quick mass transfer [34]. Similarly, encapsulation of GA-3 (gibberellic acid) was carried out using alginate–chitosan and chitosan–tripolyphosphate, and it was reported that leaf area and chlorophyll and carotenoid content increased in the latter case [61]. β-Cyclodextrin-modified magnetic graphene oxide materials were used to trace the five PGRs in a vegetable sample [62]. Similarly, a crystalline porous polymer material (Fe_3_O_4_@COF(TpDA)) was used for PGR detection in fruits and vegetables [63]. In another study on coriander and garlic growth, it was observed that graphene quantum dots were found suitable to enhance the PGR level [64]. They reported that graphene quantum dots promoted the growth rate in almost all plant parts such as roots, shoots, leaves, flowers and fruits. However, the main difficulties in the utilization of PGRs are their degradation when coming into contact with heat and light. Hence, to cope with these environmental problems, nanoscale materials are employed that promote more stability and less environmental risk [65].

### 3.3. Role of Nanotechnology in Plant Genetic Engineering

Plant genetic engineering is found to be an important tool used to improve the quality of the crop plants in terms of their yield, enhancement in secondary metabolite contents in medicinal plants, etc. *Agrobacterium*-mediated and gene-gun transformation methods are widely employed techniques in plants. However, there are some limitations to these techniques; for example, *Agrobacterium*-mediated transformation is purely host specific, whereas tissue damage may be reported in the gene-gun method. A new method based on nanoparticle-mediated genetic transformation opens the door to shaping modern agriculture systems by removing these restrictions. The NP-mediated transformation boost is a completely new strategy in plant biotechnology in promoting a sustainable agriculture system. Gene transfer and protein delivery with the use of nanotechnology have their role in crop engineering, drug delivery and environmental monitoring. Conjugation of DNA with NPs decreases the chance of degradation of DNA through DNase enzymes and ultrasound. These benefits allow transferring genetic material in plant cells by the use of ultrasound-assisted nanotechnology. It is based on the introduction of part of the exogenous nucleic acid into the target leading to the interference in the normal transcription of the target gene. NPs have the ability to deliver DNA, RNA and proteins (Figure 3A–C), and in addition to this, chloroplast-targeted transgene delivery has also been widely employed by several scientists. Among these, RNA interference (RNAi) is a widely employed technique to protect and enhance the quality of the crop. RNAi delivers double-stranded RNA or small interfering (si) RNA in the nucleus affecting the normal gene coding process [66] (Figure 3B). It is found that NP-mediated RNA delivery is most widely employed due to its high specificity and low-cost development. The applied NMs in the RNAi delivery system protect siRNA and help them to reach the target site with high efficacy [67].

Layered double hydroxide (LDH) bioclay was used as a carrier for the delivery of dsRNA (double-stranded RNA) in cowpea and tobacco for crop protection [68]. The stability of dsRNA was maintained by LDH bioclay followed by the release of dsRNA by the formation of carbonic acid on the surface of the leaf from atmospheric CO_2_ and H_2_O. Kwak et al. and Demirer et al. found that single-walled carbon nanotubes have the ability to deliver genetic material into chloroplasts and nuclei separately [69,70]. Carbon nanotubes (CNTs) can enter the plant cells through the LEEP (lipid exchange envelope penetration) process [71]. Similarly, mesoporous silica nanoparticles (MSNPs) are able to deliver genes with controlled release aided by Au NPs. Meanwhile, the loading of a gene and its chemical inducer takes place on MSN and its capping is performed by Au NPs which reduce leaching and increase efficient gene expressions [72]. A successful transfer of a gene with the help of silicon dioxide NPs was achieved in maize and tobacco. Insect-tolerant novel crop varieties are also developed using the NP-assisted delivery process. For instance, DNA-coated NPs are used to bombard cells and tissues as bullets in gene-gun technology in order to transmit the requested genes to selected plants [34]. Moreover, in tobacco plants, a DNA nanostructure was used to transfer siRNA for silencing continuously expressing the GFP gene [73]. Similarly, in *Arabidopsis* plants, PEI-coated Au NPs were used to transfer siRNA and silenced the NPR1 (nonexpressor of pathogenesis-related gene 1) gene [74]. Ideally, positively charged low-cost NMs devoid of heavy metals are best suited for the delivery of negatively charged genetic material into plant cells. The above reports show the potential of biomolecules in plant cells and explore the possible ways for the development of NP-mediated genetic transformation.

### 3.4. Role of Nanotechnology in Plant Pathogen Detection and Crop Protection

Pathogen infection of the crops is one of the important factors accountable for the reduction in crop yield. Numerous emerging contagious diseases along with the old ones reduce the continuous supply of food [75]. As per the estimation, insect pests caused a 14% loss worldwide, whereas a 13% loss was caused by weeds and plant diseases separately, and the overall value of this crop loss is USD 2000 billion per year. Reduced plant growth, low yield and poor productivity are some of the consequences of chronic stress in plants due to pathogen attacks. Generally, the symptoms can be seen on leaves, stems and fruits, on the basis of which the diagnosis of diseases could be initiated [76]. However, early diagnosis is not possible because plant diseases are initially symptomless. Therefore, early detection of disease and pathogens along with the regular monitoring of plant health should be a prerequisite process in agriculture. This approach will also help to prevent the development of other diseases. Various methods have been employed for the detection of the disease such as polymerase chain reaction (PCR)-assisted techniques of DNA hybridization [77], ELISA-based tissue print or direct dot-blot immunoassay (DTBIA) DNA-based methods and immunoassays [75]. However, the above methods have some limitations such as that they show low sensitivity, are time consuming and require a large amount of target tissue. Nanotechnology, NPs and QDs (quantum dots) have emerged as important tools for the rapid detection of pathogens with high accuracy. NPs are able to detect pathogens directly or be utilized for the detection of the compounds specific to the disease. Similarly, biosensor-based antibodies and DNA, nanoimaging and nanopore DNA sequencing have the potential to play an important role in the early recognition of pathogens with a high rate of specificity and rapid detection [78]. Moreover, nanodiagnostic kits have the potential to detect the plant pathogen and help farmers in the control and prevention of epidemic diseases. In addition to this, various nanosystems and nanodevices are being employed for the sequencing of DNA, and the use of these techniques is rapidly growing due to their flexibility and high sensitivity as compared to conventional techniques. Therefore, for rapid detection using these advanced techniques, phytopathologists and nanopathologists need to work constantly, so as to benefit the farmers. Figure 4a depicts the role of nanotechnology in plant disease control.

Nanophytopathology is a new era in plant pathogen detection and crop protection that uses nanotechnology to detect and diagnose plant pathogens at an early stage which may open doors to preventing epidemic diseases. Nowadays, plant pathologists apply the combination technique of nanophytopathology to explore plant pathogen detection and measurements. Nanophytopathology is applied to monitor the pathogen population, the interaction between the plant and microbes and the transfer of genetic material between the pathogen and the host. Figure 4b summarizes the potential role of nanophtopathology. Similarly, nanochips are a kind of microarrays that comprises fluorescent oligo capture probes through which hybridization can be detected [79]. These nanochips are well known for their ability to sense single nucleotide variations of the virus and bacterial genome [79]. To detect *Xanthomonas axonopodis* pv. *Vesicatoria*, the causal organism of bacterial spots in solanaceous crops, Yao et al. used fluorescence Si NPs along with antibodies [79]. Similarly, Singh et al. utilized nano-Au immunosensors to sense *Tilletia indica* which causes Karnal bunt in wheat [80]. The physiological condition of plants changes in response to stress conditions. Many of these defense responses are controlled by plant hormones, mainly methyl jasmonate and salicylic and jasmonic acid. Furthermore, carbon nanotubes are also used to detect pathogens in plants and the environment. To detect *Botrytis cinerea* in healthy-looking apple plants before the appearance of symptoms of the disease, carbon-nanotube-based nanosensors were utilized that helped to give information about the pathogen before the spread of the infection to the field. Hence the information derived through these sensors is also useful to design pesticides against the specific pathogen [81,82]. Another study on strawberry crop wherein ZnO NPs were applied in the visible light resulted in remarkable strawberry crops [83,84]. They found that ZnO NPs inhibited the growth of the plant pathogen (*B. cinerea*) by 12% as compared to the control plant. Similarly, the antifungal activity of Cu NPs and Ag NPs against *Alternaria brassicicola* and *Bipolaris sorokiniana* was examined, and it was found that mycelium growth was moderate on Cu NPs, whereas strong inhibition was reported in the case of Ag NPs [83]. Peptide-encapsulated SWCNTs were used for bacterial detection and classification, where 16 different peptides were used [85]. Gold NPs were applied for the detection of *Begomovirus* in chilli and tomato, and the results show that the assay was able to detect 500 ag/µL of begomoviral DNA [85]. The above report helps in broadening our understanding of early pathogen detection and monitoring plant health which will be beneficial for better crop yield and productivity.

## 4. Tailored Nanomaterials for Sustainable Agriculture

The agricultural control process can be monitored by nanotechnology, especially through its nanosize size. The potential interests such as improvement in the quantity and quality of food, reduced participation in agriculture and increased absorption of nanoscale nutrients from the soil are the main reasons for the rapidly increasing constraint. Natural resources, agriculture and food are part of competitions such as susceptibility, sustainability and human health. The purpose of nanomaterials in agriculture is to reduce the amount of dispersed chemicals, loss of nutrients in fertilization and management of the nutrients and pests to increase the yield. Nanomaterials are being tailored in a special way for agricultural use, viz., nanofertilizers, nanopesticides, nanofungicides, nanobioremediation and nanobased biosensors and absorbent materials. Figure 5 summarizes the positive effect of different nanomaterials in sustainable agriculture.

Sustainable development and food security are the major rising problems in agriculture which can be controlled by recently developed important technological advancements. These continuous agricultural advancements and the use of natural and synthetic materials play a vital role in overcoming the world’s booming population’s rising food demand. In particular, nanotechnology has the capacity to propose useful alternatives to different types of issues related to agriculture. Nanoparticles are scientifically important because the distance between bulk materials and molecular or atomic structures is negligible in this case. To save many species from extinction and conserve the environment, sustainable farming includes the possible use of fewer agrochemicals. For instance, sustainable agriculture and environmental systems are influenced by scrutiny and quality control, soil and plant health, and protection. Critical aspects of sustainable agriculture are agrochemical and organic molecule delivery systems, as well as the transfer of DNA molecules or oligonucleotides into plant cells. Recently, the slow and controlled release of fertilizers, herbicides and pesticides synthesized by nanotechnology has gained great interest in agriculture in order to assure environmentally sustainable agricultural activities [34]. In this section, we discuss the customized nanomaterials and their use in sustainable agriculture.

### 4.1. Nanofertilizers

The use of nanotechnology in agriculture could be a potential solution to the ever-increasing human hunger problem. The integration of nanotechnology with fertilizer may serve as a promising advance to combat food scarcity and environmental degradation. Nanoparticles containing nutrients, which are protected by a thin nanoscale polymeric layer, can enhance plant growth and improve the efficiency of conventional fertilizers as one or more nutrients can be delivered to the target as a nanoemulsion or nanoparticle. Furthermore, the increased surface tension by nanocoating on fertilizers allows plants to adhere to material more efficiently. The beneficial effect of nanofertilizers led to an increase in plant potential and a reduction in the negative consequences of conventional fertilizers [86]. Some approved nanofertilizers that are being used in the world today are Katra nano magnesium sulphate, prime aavirat growth booster, Iffco nano 12 manure, agro kill, carbon nano powder, nano calcium, nano capsule, nano micro nutrient, PPC nano, nano max NPK, TAG nano, Nano green, nano agro care and Biozar nanofertilizer [87]. It is known that conventional fertilizers are usually applied by foliar spray and soil dispersal. The final concentration of fertilizer that needs to reach the plant is the determining factor of the mode of application. Usually, a very small amount of conventional fertilizers affect the target site [88]. In applied conventional fertilizers, about 40–70% of nitrogen, 80–90% of phosphorous and 50–90% of potassium content is lost to the environment, causing unsustainability and economic losses [89]. These complications increase the constant use of fertilizers which adversely affect the balance of nutrients in the soil and the native flora and fauna and increase environmental pollution [88]. In recent years, the requirement for conventional fertilizers has been reduced by using nanofertilizers which increase the fertility of the soil and the quantity and quality of crops. Nanofertilizers are less harmful and non-toxic to humans, increase crop resistance to disease and drought, decrease crop involvement and increase earnings [90]. NFs are tailored nutrient fertilizers made up of a nanostructured articulation that delivers nutrients to plants either completely or partially for the assimilation or slow release of active constituents. Nanofertilizers provide the nutrients to the plant either by encapsulating inside nanomaterials and nanoporous materials or by coating with a thin polymer film. It is proved that nanofertilizers show better results compared to conventional fertilizers in fulfilling the requirements of plant roots, promoting disease resistance, enhancing plant growth, improving the stability of plants and encouraging deeper rooting of the crops [91].

### 4.2. Nanofungicides

In the field of agriculture, pathogenic fungi are the major barriers to and problems of agricultural growth, which cause more than 70% of diseases in plants and reduce the yields by up to 100% [92]. Conventionally, fungal diseases are treated by chemical fungicides which have harmful impacts on plants’ health and the surrounding environment. The use of nanofungicides is a valuable solution to overcome these problems depending on the size and shape of the NPs. The danger induced by fungicides is significant as their effects are both harmful and beneficial. NPs are predicted to be a valuable alternative to fungicides that come with phytocompounds and biocontrol agents. Some commercially available nanomaterial-based fungicides are Subdue MAXX and Cruiser MaXX [87]. Initial studies on nanofungicides were directed in early 1997 to unravel the incorporation of fungicides in solid wood [93,94]. Subsequently, conventional biocides and broad NPs with anti-fungal properties have been studied. Chitosan, polymer mixes and silica are the most commonly used nanoparticle transporters. A wide range of fungi was studied which are used as nanocarriers for nanofungicides and insecticides. Silver, TiO_2_, carbon, silica and alumino-silicates have been considered as effective antifungal agents [95]. Among them, Ag has the highest potential which increased the percent of seed germination and seedling weight [95]. Silver and TiO_2_ nanoparticles were reported to inhibit various plant pathogens. Kannan et al. reported that nanosilica treatment enhanced plant resistance by increasing phenolic compounds [96]. Besides these, Ag NPs are widely studied nanoparticles for their innumerable applications and due to their fast response delivery in bacteria and fungi [97]. Gajbhiye et al. found that AgNPs and fluconazoles have antifungal activity toward different fungi such as *Fusarium semitectum, Phoma herbarum* and *Phoma glomerata* [98]. Similarly, Ag_2_S NPs and Ag NPs have also been used against *Aspergillus niger*, *Fusarium oxysporum*, *Magnaporthe grisea, Bipolaris sorokiniana, Fusarium culmorum* and *Colletotrichum* [99,100,101,102]. Similarly, MgO and ZnO were used against *Fusarium oxysporum, Alternaria alternata, Rhizopus stolonifer* and *Mucor plumbeus* [103]. Park et al. showed the antifungal activity of Ag-silica NPs (nanocomposites) against powdery mildew infection by spraying them over pumpkin leaves for 3 days [104]. Copper also shows efficient antifungal activity in a nanocomposite with a polymer [105]. Nanofungicides are also synthesized by using algae [106]. In tomato, pepper, potato and eggplant, *Sargassum longifolium* with silver nanoparticles was used as a nanofungicide against *Fusarium* wilt which is caused by *Fusarium oxysporum* [107].

### 4.3. Nanopesticides

In commercial agriculture, the use of pesticides is a regular process, and for this purpose, new, efficient and target-specific pesticides are continuously developed. Approximately, 2 million tonnes of pesticides are utilized annually worldwide [108], and this is the reason why every year, a large number of pesticides are screened. It is also observed that a very small amount (0.1%) of pesticides used reaches the target pests while the rest (99.9%) remain in the environment which adversely affects the environment and human health [109]. Regarding their effect on non-targeted species, pesticides resulted in resistance in insects, pathogens and weeds [110]. However, it is noteworthy that if pesticides were not available in the world, they would cause a loss of life by a factor of 1000 compared to each life lost due to pesticides [111]. Biopesticides have been shown to have the potential to reduce the destructive effects of synthetic insecticides, but their sluggish and environment-dependent efficacy against pests creates a limitation in their use. To overcome these limitations, nanopesticides are the best viable alternative to conventional pesticides. The main advantage of using nanopesticides is that they have minimal effect on non-targeted insects and are eco-friendly. Moreover, nanopesticides are water-soluble components, unlike conventional hydrophobic pesticides, possess high bioactivity, are applied in small quantities and are quickly taken up by cells [112,113,114]. Some commercially available nanopesticides include nano green, nano-pole insecticides, encapsulated plant protection agent (PRIMO Maxx and Karate^®^ ZEON), insecticides such as cruiser MaXX, nano-pole insecticides, etc. In addition to these nanomaterials, copper oxide (CuO NPs), zinc oxide (ZnO NPs), magnesium hydroxide (MgOH NPs), magnesium oxide (MgO NPs) and silica dioxide (SiO_2_ NPs) are also effective against harmful pests [112,113,114].

In the presence of suitable nanomaterials, the active ingredients are gradually degraded and released in a controlled manner which can lead to effective pest control in the long run [115]. Therefore, it can be said that the use of nanopesticides is very important for the sustainable and effective management of various pests and has the potential to reduce the use of synthetic chemicals. The mode of action of nanopesticides is different from that of conventional pesticides [116], and they are transported in colloidal and dissolved states [117]. In the case of nanopesticides, mobility and degradation of active ingredients by soil-inhabiting microorganisms could be increased by the solubility of active ingredients. In other words, nanopesticides are quite efficient, revolutionary and target-specific pesticides. Recent developments in the field of nanotechnology have resulted in the creation of a new generation of pesticides. While chemical pesticides have deteriorating consequences on biodiversity and human beings by killing the non-targeted organisms, nanopesticides are transported to their targets in the form of nanospheres, nanocapsules, nanopolymers, etc. Besides the benefits of nanopesticides, there are some matters of concern such as biosafety issues in the agricultural field and their long-term effect on humans, the surrounding environment and frequently exposed workers. The nanometric size of the permitted pesticide formulations should be carefully analyzed before exposing them to the environment. For example, Europe has approved synthetic amorphous silicon dioxide as a nanomaterial in the form of stable aggregated particles of particle size > 1 μm, with primary particles of nanosize [118]. The protection of plants and the production of food by using nanomaterials is an under-explored area, and it is very helpful for the environment and humans to develop non-toxic and encouraging pesticide transport systems to increase global food production by reducing harmful environmental effects on the ecosystem [119].

### 4.4. Nanobased Biosensors

Identification of contaminants, heavy metals and toxic compounds from domestic and industrial waste sources, soil monitoring and other dynamic formations is comprehensive and exhaustive work which requires fast, consistent and low-cost systems for detection [120,121]. For a quantitative analysis of environmental samples, traditional precise and sensitive analytical systems such as chromatographic and spectroscopic techniques are used which require advanced equipment, expert staff and complex multistep sample preparation. To eliminate the issues related to these technologies, a variety of new biosensors are being developed. Biosensors are analytical tools, and some of these biosensors rely on nanotechnological tools called nanobiosensors. Nanobiosensors are a revolution in this field with great potential to solve these problems by increasing robustness, sensitivity, point of use and flexibility [122,123]. A nanobiosensor has the ability to find any biochemical and biophysical signal connected to a particular analyte [124] and to estimate its existence and concentration in water, soil and wastewater [125]. In the field of agriculture, nanobiosensors are used for the detection of pesticides, herbicides, fertilizers, toxins, heavy metals and pathogens, soil monitoring for quality and fertility and as indicators for seed viability and precision agriculture (Figure 6a) [126,127]. Heavy metals can be detected by biosensors with aptamer- and DNA-based properties which are appropriate for screening and monitoring food safety. In the case of nanofertilizer nanobiosensors, nanosensors may also be used to observe plant growth by evaluating the cross-talk between the rhizosphere and roots which leads to the development of an energetic, accurate and intellectual nanofertilizer delivery platform. Nanobiosensors are transportable, small, specific, extra sensitive, reliable and also used for real-time monitoring. On the basis of these properties, nanobiosensors have priority over current conventional sensors.

Different types of nanosensors have been explored in plants such as plasmonic nanosensors, carbon-based electrochemical nanosensors, fluorescence resonance energy transfer (FRET)-based nanosensors, nanowire nanosesnsors and antibody nanosensors. In addition to the above, different molecular methods, for example, polymerase chain reaction (PCR), real-time PCR, Raman and fluorescence spectroscopy, infrared spectroscopy, etc., exist [128]. Research has been focused on genetically encoded nanosensors or FRET-based nanosensors for improving the resource allocation efficiency for pathogens or early recognition and amplification of resource deficiency and their regulation for pathogens. Some examples of nanosensors in plant pathogen detection, plant disease monitoring and management are discussed here, such as: (1) Fluorescent silica nanoparticles (FSNPs) detect pathogens related to bacterial spot disease in tomato [79]. (2) Gold nanoparticles detect *Tilletia indica* responsible for Karnal bunt disease in wheat [80]. (3) Luminescent semiconductor nanocrystals (QD) detect beet necrotic yellow vein virus (BNYVV) responsible for rhizomania disease in beet [129]. (4) Gold nanoparticles detect tristeza virus responsible for tristeza disease in citrus [130,131].

### 4.5. Nanobioremediation

Due to rapid urbanization and colonization, there has been a significant increase in the rich industrial release of various types of wastewater which is an anthropogenetic activity that destroys soil health and air and water quality. To control the waste materials released from various sources, a large number of technologies have been developed including physical remediation, chemical remediation, phytoremediation and microbial remediation. Bacterial and fungal metabolisms also destroy various toxic substances trapped in the waste materials. For the bioremediation process of waste materials, some important bacterial communities are *Bacillus, Streptomyces, Pseudomonas, Thiobacillus, Achromobacter, Acinetobacter, Nitrobacter, Alcaligenes, Flavobacterium* and *Micrococcus*; among the fungi are *Fusarium, Penicillium, Mucor, Pleurotus, Aspergillus, Trichoderma* and white rot mushrooms, and AMF are known to be efficient organisms. In other words, bioremediation for sustainable development is an innovative method of developing waste management, and the use of microbes for this process is considered economically and environmentally safe [132]. In addition to the use of soil microbes, earthworms also play a dynamic role in bioremediation waste management [133]. Nanotechnology is also involved in bioremediation (nanobioremediation) and plays an important role (Figure 6b). In other words, nanobioremediation is a progressive and fast-developing advanced technology in which biologically synthesized nanoparticles are used to eradicate contaminants from the environment. Nanobioremediation significantly enhances the efficiency of the disinfection process. Different types of nanomaterials are used instantaneously or successively with microbes and plants, or they can be utilized as nanocarriers for microbial bioabsorbents to expedite heavy metal removal [134]. Rizwan et al. showed the main nanomaterials that are involved in nanobioremediation, for example, nanosized dendrimers, nanoiron and its derivatives, single enzyme nanoparticles, carbon nanotubes, engineered nanoparticles, etc. [135], whereas in the case of nanobioremediation, engineered polymeric nanoparticles, bioremediated soil and hydrophobic contaminants [136].

## 5. Role of NPs in Biotic and Abiotic Stress

Agriculture is the economic pillar of developing countries for the better life of human beings on a global scale [137]. In ecosystems, biotic and abiotic stresses drive climate change that damages the particular balance between the environment and food production, is related to crop productivity and crop failure and can lead to significant issues [138]. Abiotic stress (salinity, drought, heat, high light and heavy metals) causes morphological, physiological, biochemical and molecular changes, production of ROS (reactive oxygen species) and damage of membrane and alters other metabolic activities [139]. Meanwhile, biotic stress (viruses, bacteria, fungi, nematodes, parasitic plants, weeds and insects) [140] leads to changes in plant metabolism and physiological damage, leading to decreased crop production. In order to reduce the harmful effects of biotic and abiotic stresses, several approaches to plant metabolism have been applied, including the use of nanomaterials. Table 1 and Table 2 and Figure 7 depict the role of nanomaterials in improving plant growth under biotic and abiotic stress. Nanomaterials are gaining global attention nowadays for protecting plants from biotic and abiotic stresses. For example, to enhance stress tolerance, plants employ key strategies such as upregulation of functional and structural protectants including compatible solutes (osmolytes) and antioxidants [141]. It is known that a low concentration of ROS acts as a signal that promotes the growth, development and defense mechanisms of the plant, but over-accumulation of ROS under stress conditions causes damage to cell membranes, DNA, proteins and other cell components which results in the inhibition of plant growth. Nanomaterials containing antioxidant enzyme activities enhance the ability of plants to scavenge ROS which improves the resistance of the plant to abiotic stress and therefore increases the yield. On the other hand, plants developed an advanced immune system to cope with biotic stresses. In plants, the first line of defense is passive. To prevent the entry of insects or pathogens into the plants, physical barriers such as cuticles, trichrome and waxes are found. Chemical compounds are also produced by plants in response to herbivory and pathogen infection [142]. Moreover, plants also have two levels of pathogen recognition that activate the defense system. Nanomaterials are also used to nullify the effect of biotic stresses. For example, Ag NPs biosynthesized by *Pseudomonas poae* strain CO showed antifungal activity against *Fusarium graminearum*, and it reduced the mycelia growth, spore germination and germ tube length, as well as severely damaged the cell wall at a remarkable level, and therefore, the production of fungal mycotoxins could be diminished by exposure to Ag NPs [143].

### 5.1. Mode of Action of NPs in Combating the Stress

The growing threat of climate change, rapid urbanization, growing population and shrinking agricultural land have increased concern about food security. Nanotechnology offers an opportunity to address various challenges faced by the agriculture sector today. However, the central focus of nanotechnology has been on increasing crop production with minimal impact on ecological sustainability. A proposed mechanism of action of NPs is given in Figure 8. The small size of the nanomaterial is essential for crossing biological membranes and transporting the drug to the targeted site in plants. The size and concentration of a particular nanoparticle are vital for the tolerance against abiotic stresses. Zinc oxide (ZnO) NPs have been applied at various concentrations to overcome Zn deficiency in plants and alleviate the toxic effect of superoxide radicals [183]. The application of ZnO NPs to *Gossypium hirsutum* increases the activity of POX and SOD while subsequently decreasing the lipid peroxidation rate. Similarly, ZnO NPs come in various shapes and sizes such as spherical (38 nm), floral (59 nm) and rod-like (>500 nm); however, the most effective were observed to be spherical ZnO NPs of a size less than 40 nm which reportedly enhances the antioxidant response against oxidative stress in the case of soybean. The pretreatment of TiO2 and ZnO nanoparticles results in a significant enhancement in SOD and GPX activity against extreme temperature stress and stabilization of the membrane by reducing H_2_O in wheat plants [184]. Similarly, a significant increase in APX, GPX, CAT and GR was reported in *Brassica juncea* when exposed to gold nanoparticles (GNPs), besides increasing the content of proline to a greater amount [185].

#### 5.1.1. Salt Stress

In the age of climate change, salinity stress is posing a serious threat to agriculture, significantly reducing overall crop production with both primary and secondary effects. Primary effects include osmotic and ionic disturbances while secondary effects include disturbances in hormonal balances, oxidative stress and nutrient imbalance [186]. Various evidence suggests that the supplementation of different types of nanomaterials substantially attenuates various types of injuries caused by salt stress and, therefore, holds promise for an adaptive adjustment. Nanotechnology is an emerging field of science and has attracted attention over the years because of its potential to promote sustainability in agriculture. Furthermore, an excess intake of sodium ions causes disturbances in various intracellular processes, particularly the generation of reactive oxygen species (ROS) that damage membrane structure and disturb metabolic pathways, which in turn deplete the cell’s energy pool [187]. Various plant growth parameters have often shown negative effects on exposure to salt stress such as decreased leaf area, chlorophyll content, gas exchange characteristics and photosynthetic pigments [188]. Ag NPs are emerging as a very important nanomaterial as it has wide applicability in various fields of science which is attributed to its small size with physicochemical properties [189]. It has been reported that Ag NPs have antibacterial and antifungal properties and can potentially be used for wastewater treatment. Khan et al. reported that treatment of seeds with silver nanoparticles before sowing reduced the ratio of Na^+^/K^+^ while increasing the activity of antioxidants [189]. Wahid et al. reported that the combination of Ag NPs and NaCl reduced hydrogen peroxide (H_2_O_2_) and electrolyte leakages [190]. It has been reported that nanoparticles efficiently transport nutrients to various locations of the plant [191]. Rizwan et al. reported that Ag NPs may be acting as agents in mediating the interaction of various nutrients with key metabolic processes, therefore promoting the growth of a plant [192]. Chlorophyll as a vital component of photosynthesis has been reported to reduce the overall photosynthetic activity under salt stress; however, silver nanoparticles have been reported to enhance photosynthetic activity at lower concentrations [193]. Similarly, in the case of pearl millet, the application of Ag NPs significantly increases the antioxidant activities of superoxide dismutase (SOD), CAT (catalase) and POD (peroxidase) [194]. Pre-treatment of seeds with Ag NPs at 100–150 nm concentration significantly improved the activity of SOD, CAT and POD under salt stress conditions in the roots and shoots of the pearl millet [194]. Oxygen radicals are reported to be scavenged by SOD and converted further into H_2_O_2_; thereafter, the enzymes such as CAT and POD break down the H_2_O_2_ into water and oxygen molecules. CAT might have a direct link to the developmental phenomenon of a plant under salinity stress due to its significant ability to scavenge ROS [195]. Moreover, the oxygen radical outburst may be a consequence of salt stress; however, increasing the activity of antioxidants could be related to the nanoparticles to reduce the stress caused by ROS [192]. Similarly, smaller-size Ag NPs have shown lower CAT activity, and larger-size Ag NPs have shown increased activity of CAT, and this behavior could be attributed to the size, nature and type of the plant species exposed to nanoparticles [196]. More importantly, one study has revealed that a lower Na^+^ and Cl^-^ concentration has been reported in various organs of the plant when subjected to Ag NP exposure. It can be assumed that Ag NPs promote the absorbance of nutrients in different locations of a plant [197]. Due to the natural ability of silicon to alleviate various kinds of abiotic stresses in plants, research has been conducted to study the effect of SiO_2_ NPs on the “Valencia” sweet orange under salt stress conditions. NaCl exposure to the “Valencia” sweet orange at 60 and 120 mM treatment significantly reduced the relative water capacity of a leaf and membrane damage which is believed to be caused by the leakage of electrolytes [191]. The exogenous application of SiO2 NPs has significantly increased the chlorophyll content and root length of Valencia plants suggesting a decrease in osmotic stress [198].

The SOS pathway plays an important role in the regulation of Na+ transportation from roots to the other parts of the plant [48]. Subsequently, it also aids in maintaining the ionic balance of the cell and thus endows the plant with salt tolerance [199]. It has been reported that the expression of CsSOS1, CsSOS2 and CsSOS3 genes was upregulated upon the exposure of plants to Si NPs, and therefore, a decrease in Na+ was reported in leaf and root tissues. Moreover, aquaporin (AQP) isoforms have a vital role in regulating the uptake of water by the roots in various plant species, and under salt stress conditions, aquaporin proteins decrease the water potential of the cell and therefore facilitate water uptake in order to mitigate the impact of salt stress [200]. The plasma membrane intrinsic protein (PIP) family plays an important role in mediating the intercellular transport of soluble solutes. The transcription regulation of CsPIP1 and CsPIP2 is upregulated by Si [201].

#### 5.1.2. Drought Stress

As stated above, abiotic stresses are limiting crop productivity worldwide and therefore pose a major challenge to global food security [202]. Drought stress has significantly affected agricultural yields in many arid regions. It has been reported that silica NP treatments of hawthorns (*Crataegus* sp.) have significantly increased their resistance level against various abiotic stresses. In addition, changes have been observed in various physiological and biochemical parameters such as malondialdehyde content (MDA), relative water content (RWC) and chlorophyll, carotenoid, proline and carbohydrate content upon the treatment of hawthorn seedlings with a series of silica nanoparticles. Chitosan nanoparticles (CSNPs) have been reported to increase stomatal conductance in both stressed and non-stressed plants. Stomatal closure through ABA signaling caused by CSNPs has also been reported in some species, whereas in some species, higher stomatal conductance was reported. This fact has been compared with the sensitivity of a plant to water stress. Furthermore, the stress activates ROS which causes lipid oxidative damage and membrane injury. Disruption in membrane integrity is often observed in plants subjected to drought stress indicated by MDA content and free radicals [203]. Drought stress has been reported to cause membrane deterioration in a large number of species [204]. The application of CSNPs has been reported to increase drought tolerance by considerably reducing the H_2_O_2_ and MDA content under drought stress in *C. roseus* leaves. Therefore, CSNPs are reported to maintain the membrane integrity and functions of the cell under drought stress. Chitosan positively regulates osmotic pressure and therefore significantly reduces the ROS-led adverse effect on the cell. Similarly, HANP (hydroxylapatite nanoparticle)-treated seeds of jute have shown significant tolerance to drought stress by activating the proline biosynthesis pathway [205]. Drought stress severely impacts the growth of corn seedlings; however, the treatment of yttrium-doped Fe_2_O_3_ NPs was reported to significantly improve the photosynthetic activity and chlorophyll content of the plant [144]. ZnO is reported to improve the seed germination rate by increasing the activity of gibberellin hormones. Similarly, Fe_2_O_3_ has been reported to cause an increase in drought stress tolerance in plants by modifying their carbohydrate metabolism and stomatal movement. The ZnO NP treatment of maize plants has been reported to downregulate the degradation of photosynthetic pigments and enhance photosynthesis. Therefore, ZnO NPs act as a potential agent for improving the drought tolerance in a number of plant species. Similarly, CuO NP treatment in maize has significantly reduced the ROS content and positively regulates the pigment system under drought stress conditions [206].

#### 5.1.3. Extreme Temperature

Temperature above a threshold level is called heat stress and below a threshold level is known as cold stress. The heat and cold stress is believed to cause significant ion imbalance in the cell and create a serious hindrance in the growth and development of the plant. The extreme temperature becomes a source of ROS which causes severe damage to the physiological and biochemical activity of a cell, besides damaging the most vital proteins—heat shock proteins (HSPs) [207]. Selenium nanoparticles were reported to combat high temperature stress. Djanaguiraman et al. reported that treatment of sorghum with selenium NPs activates the antioxidant machinery to scavenge the ROS produced by high temperature stress, therefore mitigating the heat stress effect [208]. Similarly, selenium NPs were reported to improve both cold and high temperature tolerance in *Lycopersicum esculentum* [209]. Similarly, photosynthetic machinery in wheat plants was reported to be affected by heat stress; however, the application of AgNPs imparts tolerance against heat stress and significantly improves various growth parameters such as root length, shoot length and fresh and dry weight [210].

#### 5.1.4. Metal Stress

Since industrialization, metal toxicity has percolated in agricultural fields and has severely reduced the overall yield and production of the crop. The metal stress is reported to cause a decrease in enzyme activities, and ion imbalance leads to the deficiency of some key nutrient elements. Furthermore, it was reported that metal accumulation in plants generates excess ROS which cause damage to the structure of the cell membrane and destroy some important enzymes [211]. However, plants possess a wide range of defense systems to tackle abiotic stresses. Moreover, the accumulation of metal ions by plants causes the formation of metal chelates which significantly restricts the efflux of metal ions, sequestration of metals and activation of the antioxidant defense system. Nanoparticles have been reported to cause alleviation of metal-induced toxicity [212]. Being small in size, nanoparticles easily penetrate into various parts of the plant and contain a strong affinity toward metal ions. Similarly, quantum dots have been reported to reduce the availability of lead (Pb) and act as an additional barrier to the cell wall. It has been further reported that if a metal ion manages to enter the plant cell, the plants counteract it by accumulating various biomolecules and nutrients in order to establish a defense system to mitigate the stress effect. Nano-TiO_2_ is believed to mitigate various harmful impacts of abiotic stress; it also significantly restricts cadmium toxicity and enhances photosynthesis in plants [213].

## 6. Plant–Nanoparticle Interactions

The interaction of NPs or ENMs with plants or biological systems may be chemical or mechanical, and the interaction depends upon various factors such as the nature and size of NPs, the physiological status of the plant and NP interaction with the environment [214,215]. The smaller size of NPs with a large surface area including catalytic reactivity is the main reason for interaction [214]. It is now well established that the size of the NPs is the main barrier to the entry of NPs into the plant cell as per the report that the maximum dimension of NPs that plants allow to accumulate in cells is typically 40–50 nm [216,217]. In addition to this, the chemical composition and type of NPs, morphology, coating of the NP surface, etc., greatly impact the absorption and accumulation by the plant [218,219,220]. Moreover, NPs also interact with other environmental components, for example, humic acids, organic matters, salt ions and biological organisms (bacteria, fungi, etc.) that affect NP uptake by plants [183,221].

Exploring the nature of interactions between NPs and plants is critical, and information is limited. The US Environment Protection Agency [222] has first suggested evaluating the effect of NPs on plant metabolism based on phenotypic characteristics, for example, germination of the seed and elongation of the root. Later, it was found that these standards were very specific to plant type and NP properties, lacking sensitivity during the evaluation of NP toxicity to terrestrial plant species. The action mechanism of NPs in the cell system is a complex and less explored area. However, various biochemical markers (e.g., metabolite composition, membrane integrity and enzyme activity) are being utilized from time to time to understand the plant–NP interaction. There are two ways (apoplastic and symplastic) that can be taken by NPs when penetrating into the plant tissue. Apoplastic movement takes place via extracellular space (outside the plasma membrane) and symplastic via plasmodesmata [223,224]. The apoplastic pathways lead to the entry of NPs inside the root central cylinder followed by movement through the xylem to the aerial part of the plant [225,226]. Certainly, the Casparian strip acts as a barrier that allows symplastic movement of the NPs via endodermal cells; moreover, restricted movement and accumulation of some NPs at the Casparian strips also take place [226,227,228]. Symplastic movement also allows NPs to move through sieve tube elements in the phloem that lead the distribution toward non-photosynthetic tissue [219,225]. In the case of symplastic movement, NPs must be internalized by the plant cell and cross the plasma membrane. There are various ways available for NPs to achieve this, such as endocytosis (Etxeberria et al. 2006), via carrier proteins [229], pore formation [71], plasmodesmata [224] and ion channels [218,230]. In another method, for example, the foliar application of NPs, they need to cross the barrier of the cuticle, and thereafter, two pathways (lipophilic and hydrophilic) mediate the mode of transport of NPs [231]. Lipophilic pathways mediate the transport through cuticular waxes following diffusion, while the case of hydrophilic pathways involves movement through polar aqueous pores present in cuticles or stomata [232]. Due to the size exclusion limit above 10 nm, the stomatal pathway appears as the most likely route for NP penetration. The movement of NPs within the plant is an important indicator that can reveal its location or accumulation in the plant. NPs transported mainly through the xylem are likely to move in an upward direction, for example, from the root to the aerial part. While in the case of phloem, NPs accumulate in plant organs (such as fruits, grains, flowers and young leaves) acting as sinks and must be applied through foliar application. However, lateral movement of NPs between the xylem and phloem is also possible [233]. In addition, the types of NPs and the plant species also play a role in influencing the transfer and accumulation of NPs [234,235]. Overall, the above information provides an idea in deciding the role of NMs we want to test.

## 7. Toxicological Concerns of Nanomaterials

It is evident from the above reports that many nanomaterial-based products are being used commercially. Day by day, many of these products are being engineered for better use. In addition, the use of these ENMs at the industrial level is increasing, and consequently, an increasing concentration of these materials is reaching the environment. Most of these ENMs are finally disposed of in the soil in uncontrolled quantities [236]. The tailored nanomaterials, for example, nanofertilizers, nanopesticides and nanobiosensors, are being utilized in agriculture systems leading to their accumulation in the soil. The accumulation of these affects soil health, soil microbiome and, most importantly, plant health [30]. Moreover, nitrogen fixation, mineralization and activities responsible for plant growth and development are also affected [30]. Some ENMs such as silver, SiO_2_, titania, iron oxides, zinc oxides, alumina, etc., are being discarded into the soil in large quantities without evaluating the risk [30]. As per the reports, ENMs are released into the environment at several stages such as the synthesis, manufacturing, usage and disposal of the products [237]. Figure 9 explains the possible ways of ENM accumulation in the soil. ENMs in the soil are accumulated directly or indirectly, via different routes, making it essential to examine the impact of this ENM accumulation in the soil, changes in their concentration and the life span of ENMs in the soil. So far, many studies have been published dealing with the impact of ENMs on soil, soil microbiome and plants, and almost no effect was observed when ENMs were used in lower doses. In one study, a positive effect of low doses of ENMs on plant growth was reported, and hence, exact doses should be taken into consideration for a real evaluation of ENM toxicity or growth-promoting activity. Overall, the regulation of the release of ENMs at various stages as shown in Figure 9 is mandatory [30]. Therefore, a careful evaluation of ENM accumulation in the environment and its risk to the plant microbiome is extremely desirable. It will enhance the wise and sustainable use of nanotechnology in agriculture systems.

## 8. Conclusions

Nanotechnology has great potential to improve the agricultural system by protecting plants from various environmental stresses, improving plant health and increasing agricultural yields. The above can be achieved by the application of nanoparticles and/or tailored nanomaterials in plants as a sustainable approach to make this field a multi-billion dollar industry. Currently, various nanobased products such as nanofertilizers (nano ultrafertilizer, nano capsule, nano max NPK, TAG NANO, etc.), fungicides and insecticides (Subdue MAXX, Cruiser MaXX, nano-pole insecticides, etc.), nanopesticides (nano green, encapsulated plant protection agent such as PRIMO Maxx, Karate^®^ Zeon, etc.) and different types of nanosensors are being utilized in the market. The industry is facing difficulties in market access, and this may be due to high production costs and the need for high volumes of nanotechnology products in the agriculture sector. However, there are many reports available supporting these green sustainable products. Nevertheless, there are still some unanswered questions that need to be clarified. Further research is therefore needed to elucidate the interactions of nanomaterials with biological macromolecules and their effects on nanomaterial toxicity, the ecosystem and health outcomes.

## Figures and Tables

**Figure 1 nanomaterials-12-02667-f001:**
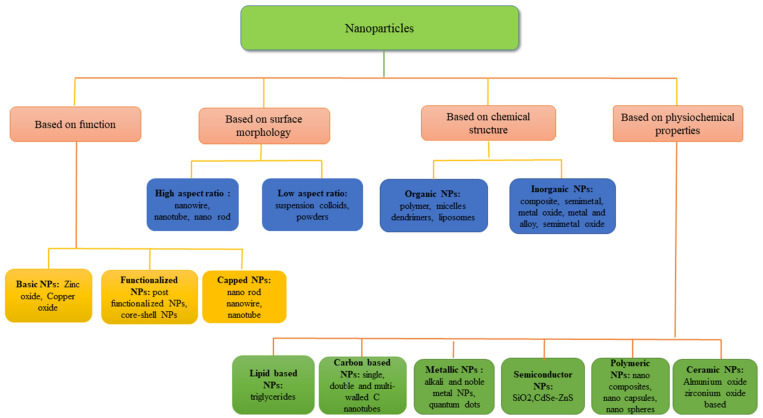
Types of nanoparticles.

**Figure 2 nanomaterials-12-02667-f002:**
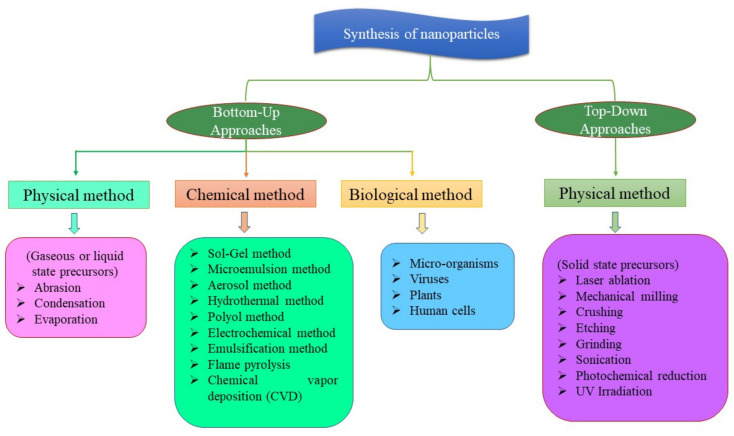
Synthesis of nanomaterials.

**Figure 3 nanomaterials-12-02667-f003:**
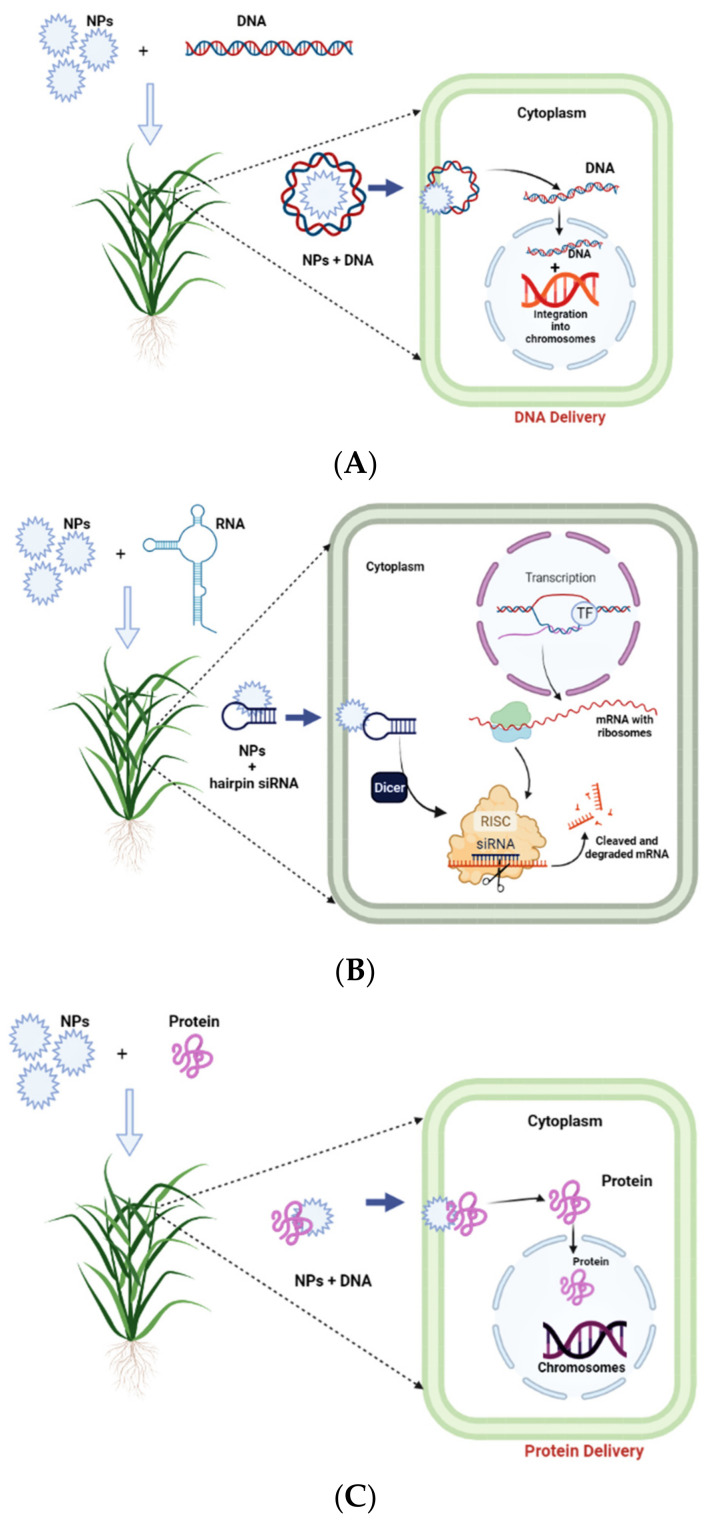
(**A**) DNA, RNA (**B**) and protein (**C**) delivery using nanoparticles.

**Figure 4 nanomaterials-12-02667-f004:**
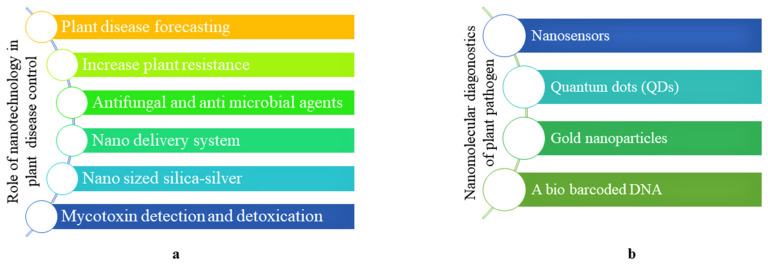
Application of nanotechnology in plant pathology: (**a**) plant disease control, (**b**) detection of plant pathogens.

**Figure 5 nanomaterials-12-02667-f005:**
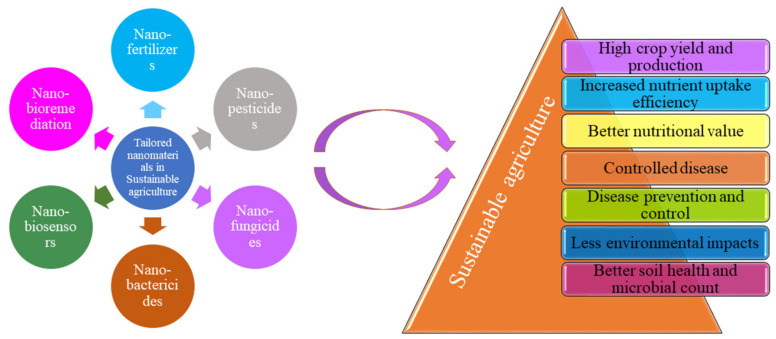
Positive effect of different nanomaterials in sustainable agriculture.

**Figure 6 nanomaterials-12-02667-f006:**
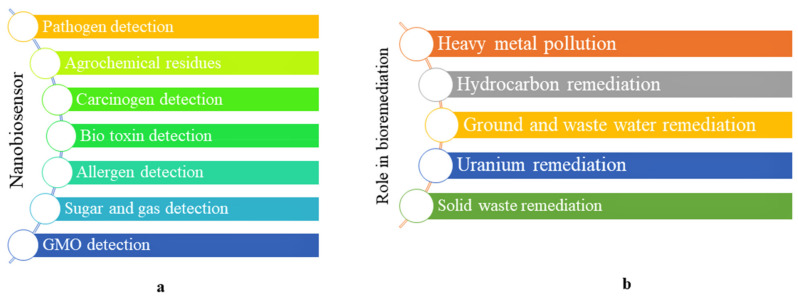
Application of nanotechnology: (**a**) role of nanobiosensors, (**b**) role in bioremediation.

**Figure 7 nanomaterials-12-02667-f007:**
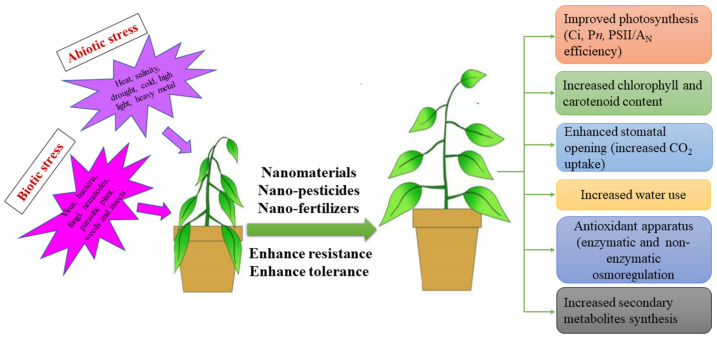
Role of nanomaterials in improving plant growth under biotic and abiotic stress.

**Figure 8 nanomaterials-12-02667-f008:**
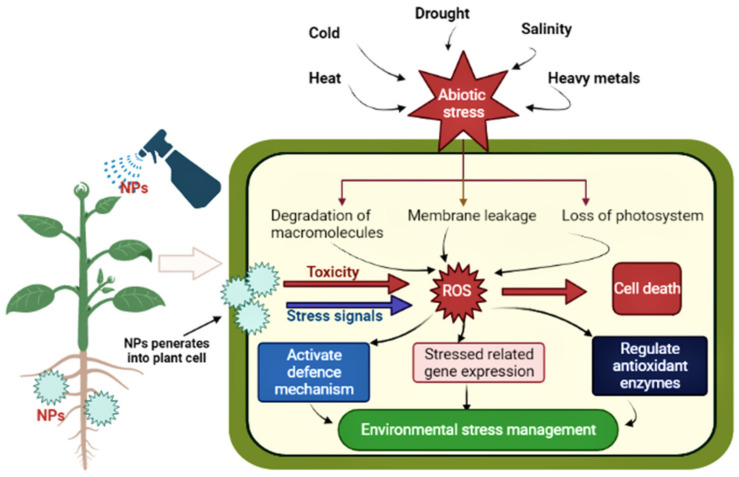
A proposed mechanism for nanoparticle-mediated abiotic stress management.

**Figure 9 nanomaterials-12-02667-f009:**
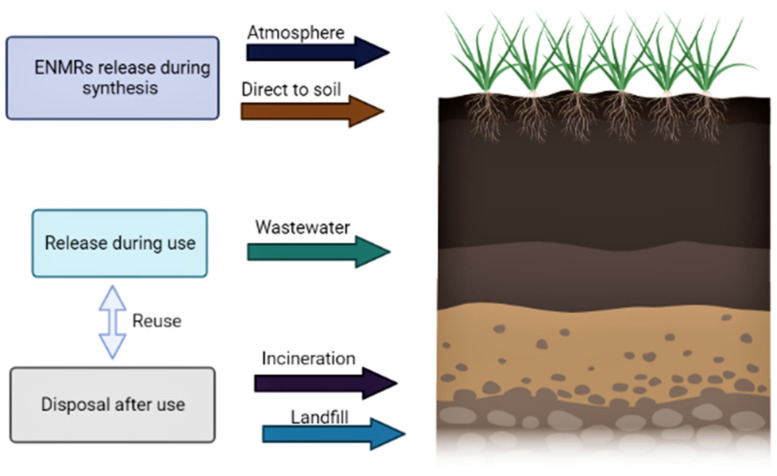
Possible ways of ENM accumulation in environment [30].

**Table 1 nanomaterials-12-02667-t001:** Effect of different nanoparticle applications on some plant species growing under abiotic stress conditions.

Name of Plant Species	Nanoparticles	Size	Concentration	Stress Type	Response	References
*Zea mays* L.	Cu	30–40 nm	3.33, 4.44 and 5.55 mg L^−1^	Drought	Higher biomass grain yield	[144]
*Oryza sativa* L.	ZnO	30 nm	50 mg L^−1^	Chilling	Regulated the antioxidative system and chilling response transcription factors	[145]
*Solanum melongena* L.	ZnO	-	50 and 100 ppm	Drought stress	Improved growth characteristics and increased fruit yield	[146]
*Zea mays* L.	TiO_2_	10–25 nm	60 ppm	Salinity stress	Enhancement of seed vigor, leaf water status and antioxidant enzyme activities	[147]
*Triticum aestivum* L.	ZnO	20–30 nm	25, 50 and 100 mgL^−1^	Drought and cadmium	Enhancement of growth, chlorophyll content, SOD and POX activities	[148]
*Trigonella foenum-graecum*	ZnO	10–30 nm	0, 1000, and 3000 ppm	Salinity	Upregulation of protein and proline levels, enhancement of the antioxidants activities, reduction in H_2_O_2_ and MDA levels	[149]
*Dracocephalum* *moldavica*	TiO_2_ NPs	20–30 Nm	0, 50, 100 and 200 mg L^−1^	Salinity stress	Improved agronomic traits and increased antioxidant enzyme activity, increased essential oil content under 100 mg L^−1^ TiO_2_	[150]
*Triticum* *aestivum* L.	TiO_2_	_	500, 1000, and 2000 mg kg^−1^	Drought stress	Improved growth, antioxidant system and photosynthetic performance	[5]
*Zea mays* L.	ZnO	37.7 ± 15.5 nm	100 mg L^−1^	Drought	Enhanced melatonin synthesis and metabolism	[133]
*Capsicum annuum* L.	Manganese	_	0.1, 0.5, 1 mgL^−1^	Salinity	Controlled salinity-modulated molecular responses	[151]
*Arundinaria pygmaea*	Silicon dioxide	20 nm	100 μM	Heavy metal	Increased protective enzymes, chlorophyll content and fluorescence, as well as plant biomass and shoot length	[152]
*Glycine max*	Ag NP	15 nm	5 ppm	Flooding stress	Enhancement of root length/weight and hypocotyl length/weight of soybean	[153]
*Abelmoschus esculentus* L.	ZnO	16–35 nm	10 mgL^−1^	Salt stress	Enhancement of the contents of the photosynthetic pigments, activity of both SOD and CAT, lowered accumulation of proline and total soluble sugar	[154]
*Mangifera indica*	Zinc oxide and silicone	nZnO < 100 nm Si = 5–15 nm	ZnO (50, 100, and 150 mgL^−1^) Si (150 and 300 mgL^−1^)	Salt stress	Improved resistance mechanism and annual productivity	[155]
*Musa acuminata*	Silicon nanoparticles	_	0, 200, 400 and 600 mgL^−1^	Salinity stress	Mitigated oxidative stress of in vitro derived plant	[156]
*Dracocephalum* *moldavica*	TiO_2_ NPs	70 –90 nm	0, 50, 100 and 200 mg L^−1^	Salinity stress	Promoted growth and ameliorated salinity stress effects on essential oil profile and biochemical attributes	[150]
*Hordeum vulgare*	Silicon	_	125, 250 mgL^−1^	Drought stress	Modified the plant morpho-physiological and antioxidative attributes and synthesis of specific metabolites	[157]
*Zea mays* L.	TiO_2_ NP	_	0, 100, 250 mgL^−1^	Cd stress	Increased superoxide dismutase (SOD) and glutathione S-transferase (GST) activities	[158]
*Glycine max*	SwCNTs	_	100 mL	Drought stress	Enhanced drought tolerance during germination	[159]
*Triticum aestivum*	Si NP	20–30 nm	1.66 mM	Heat stress	Restoration of the heat-stress-provoked ultrastructure-l distortions of chloroplast and nucleus, enhanced photochemical efficiency of the photosystem II	[160]
*Zea mays* L.	Si NPs	_	0, 300, 600, 900, 1200 mg L^−1^	Cadmium stress	Early growth and enhanced physio-biochemical and metabolic profiles of fragrant	[161]
*Ocimum basilicum* L.	TiO_2_	_	_	Drought stress	Modulated toxic effects, improved biomass accumulation and RWC	[162]
*Lycopersicum esculentum*	SiO_2_	_	1–2 mM	Salinity stress	Increased root growth, weight, seed germination	[163]

**Table 2 nanomaterials-12-02667-t002:** Effect of different nanoparticle applications on some plant species growing under biotic stress conditions.

Name of Plant Species	Nanoparticles	Size	Concentration	Stress Types	Response	References
*Bougainvillea*	CuO NPs	5–20 nm	80 and 100 ppm	*A. niger*	Antifungal	[164]
*Malus species*	CuO NPs	80 nm	0.05–1 mg mL^−1^	*Alternaria mali, Diplodia seriata*, *Botryosphaeria dothidea*	Antifungal	[165]
*Malus species*	ZnO NPs	52–70 nm	0.05–1 mg mL^−1^	*A. mali, Botryosphaeria dothidea*, *D. seriataby*	Antifungal	[166]
*Nicotiana benthamiana*	Fe_3_O_4_ NPs	20 nm	100 μg mL^−1^	*Tobacco mosaic virus* (TMV)	Antiviral	[167]
*Triticum aestivum* L.	Ag NP	19.8–44.9 nm	5–20 μg mL^−1^	*F. graminearum*	Antifungal	[143]
*Triticum aestivum* L.	TiO_2_ NP	<15 nm	25, 50, 75 μL from 0.1 mg mL^−1^	*Puccinia triticina*	Antifungal	[168]
*Lycopersicum esculentum*	Au NPs–chitosan, C-NP	80 nm	25–75 μg mL^−1^	*F. oxysporum*	Antifungal	[169]
*Saccharum officinarum*	ZnO NPs Zn NPs	72–183 nm	3–20 ppm	*Holotrichia sp*	Insecticidal	[170]
*Oryza sativa*	Ag NPs	100–250 nm	_	*R. solani, F. moniliforme*	Antifungal	[171]
*Gossypium sp.*	Ag NPs	63–85 nm	1 mM	*Earias insulana*	Insecticide	[172]
Unidentified plant	Ag NPs, Au NPs	8–510 nm	4.5 mM AgNO_3_ 5 mM gold	*S. nidulans, Trichaptum biforme, P. italicum, F. oxysporum, Colletotrichum gloeosporioides, Pseudomonas aeruginosa, Aeromonas hydrophila, Escherichia coli, Citrobacter freundii, Listeria monocytogenes, Staphylococcus epidermidis*	Antifungal/Antibacterial	[173]
*Pongamia pinnata*	Ag NPs	10–25 nm	0.0062–1.6 mg mL^−1^	*P. ultimum*	Antifungal	[174]
*Phyllanthus emblica*	Ag NPs	19.8–92.8 nm	5–30 μg mL^−1^	*A. oryzae*	Antibacterial	[175]
*Oryza sativa*	Chitosan–Fe_2_O_3_ NPs	50–70 nm	0.25–1%	*R. oryzae*	Antifungal	[176]
*Oryza sativa*	ZnO NPs	40.5–124 nm	4–16 μg mL^−1^	*X. oryzae pv.* *oryzae*	Antibacterial	[177]
*Mangifera indica*	SNPs were synthesized by lemon plant leaves	_	20–160 ppm	*Bactrocera zonata*	Insecticide	[178]
*Gossypium* sp.	TiO_2_ NPs	95 nm	31.25–1000 ppm	*Spodoptera littoralis*	Insecticide	[179]
*Triticum aestivum*	MWCNTs	_	62.5–500 μg mL^−1^	*F. graminearum*	Antifungal	[180]
*Ricinus communis* (Linn)	Ag NPs	_	10^3^ M	*Pericallia Ricini*	Insecticide	[181]
*Solanum tuberosum* *Lycopersicon esculentum* *Malus* *Domestic*	TiO_2_ NPs	_	0.8 mg plate^−1^	*F. solani* *Venturia inaequalis*	Antifungal	[182]

## Data Availability

Not applicable.

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
