# Peer review of "Nanotechnological Interventions in Agriculture"

_nanomaterials, 2022, doi:10.3390/nano12152667_

Round 1
Reviewer 1 Report
It covers a wide range of fields, including the effects of metal-based nanoparticles on pathogens and pests, the effects of nano-fertilizers, and the effects on genetic information. I could read it very interestingly. There are some points such as TiO2, where 2 is not a subscript. Please revise.
Author Response
There are some points such as TiO2, where 2 is not a subscript. Please revise.
Explanation: All the compounds including TiO2 have been revised and highlighted throughout the manuscript.

Reviewer 2 Report
See the file attached.

Author Response
Response to reviewer#2
The paper reviewed is quite interesting, a brief review on agronanochemicals, especially valuable for those not familiar with the agricultural application of nanomaterials. The Authors gathered and summarized numerous papers and results, making their review a valuable start‐point for further reading and searching for information. However, I have some comments:
- Nanotechnologists have promised world‐saving products since years. Do the Authors know any commercialized, commercially available and field‐used product (fertilizer, plant protection agent, sensor system) based on nanoparticles (or other nanomaterials), or is agronanotechnology still the future? If there are such products on the global market, it is a valuable information to be added to the conclusions of this review.
Explanation: We are agree with the comments and suggestions. We have added the information as requested. A. We have added the name of some approved nanofertilizer used in the world today for example; Nano Ultra-fertilizer, Nano capsule, Nano max NPK, TAG NANO, Iffco nano 12 manure, Prie Aavirat growth booster, agro kill etc. A complete list has been added in the section 4.1.
- Fungicide such as Subdue MAXX, Cruiser MaXX etc.
- C. Insecticide such as Cruiser MaXX, Nano-pole insecticides, nanoemulsion etc.
- D. Plant protection agent such as PRIMO MAxx and Karate® ZEON etc.
- Nanopesticides such as Nano green, encapsulated plant protection agent, Agro nanotechnology crop etc. Along with these nanoparticle such as silver, aluminum oxide, zinc oxide, magnesium hydroxide etc. has also been found effective against pest.
- The detail description of role of nanosensor in pathogen detection, plant disease monitoring and management with example has also been added to the section
The major details of the above example has been added in their corresponding heading sections. Along with this some information has also been added to the conclusion part.
- Nanomaterials' long‐term ecological and toxicological effects and their environmental durability and transport via trophic chains are still poorly recognized. This does not let you forget about the risks of the wide usage of these chemicals. A brief discussion on the hazards of introducing nanochemicals to the environment should be
Explanation: We are totally agree with the reviewer comments. Indeed the topic is very important and it’s required a special attention. Many of the engineered materials are finally disposed directly into the soil that have several impacts including effects on soil microbiome and plant health. Considering the length of the manuscript we refrain from including this topic as the MS becomes longer. However, a brief discussion has been added on this topic at the end of the manuscript in the form of a new section 6.0. A new figure (figure 9) has also been added to this section.
- The chemical formulas must be carefully checked throughout the manuscript – in many places numbers of atoms are not subscripts (e.g. "TiO2", "CeO2" on page 3; "ZnSO4" on page 8, "TiO2 and SiO2" on page 7), or charges are not subscripts (e.g. "Na+/K+", "Na+" on pages 8 & 20).
Explanation: Corrected throughout the manuscript.
- The Authors should standardize the usage on "nano‐ "prefix. Now there is "nano‐ phytopathology", "nanophytopathology", "nano‐ chips", "nanoparticles", "nano rods", "nano‐fertilizers", "nano‐based", "nano‐ fungicides" I suggest form without space or dash: nanorods, nanochips, nanofertilizer, nanopesticide etc.
Explanation: Corrected throughout the manuscript.
- The usage of plant names should also be unified. Sometimes the Authors have used only Latin names, sometimes both Latin and English, while in other places – only English I suggest using both Latin and English in the place of the first usage of the name and English in later parts of the text.
Explanation: We have tried to present the scientific name of the plant throughout the manuscript.
- In numerous lines, double‐space must be deleted (e.g. "by a virus, bacteria and fungi" on page 8), or missed space should be added (e.g. "siliconmediated" on page 42).
Explanation: Corrected.
- Flame pyrolysis, CVD and electrolysis are chemical methods. Moreover, electrolysis is an electrochemical method (Figure 2).
Explanation: Figure 2 has been changed. Flame pyrolysis and CVD have been described under chemical method. Electrolysis has been removed from the picture because electrochemical method has already been mentioned under chemical method.
- In [60], the ZnO NPs coated with phytomolecules have been studied (not phytomolecules covered with ZnO!).
Explanation: Corrected.
- "branchemakeologies", "plantsmicroorganismsganism" (page 7) ‐???
Explanation: Corrected as “repeatedly branching chains”, “plants microorganism”.
- 3 – there is A, B and D. Where is 3C?
Explanation: Typo error. There is no figure d. It is corrected as 3c.
- Abbreviation of Latin names is possible if the full name has been given Therefore on page 13 should be "To detect Botrytis cinerea in" and "of plant pathogen (B. cinerea)". For "A. brassicicola" (page 13), "L. esculentum" (page 21), full names should be used.
Explanation: Full name has been added.
- Many abbreviations have been used without explanation, g. GA‐3, dsRNA, SOD, CAT, POD, and ABA.
Explanation: GA‐3, dsRNA, SOD, CAT, POD explained in first appearance. ABA explained (page 9).
- Is notation “of CsPIP1; 1 and CsPIP2; 2” correct (page 20)? Maybe it should be “of CsPIP1 and CsPIP2”?
Explanation: corrected as CsPIP1 and CsPIP2.
- Page 16: "more than 1 million pesticides were estimated in 2009". No! In the cited monograph, there is information that 1 million species of insects are known.
Explanation: According to the new report, the sentence has been improved. The reference to the claim has been added to the reference list in place of the old reference.
- I suggest the usage of another abbreviation for hydroxyapatite nanoparticles because “CaNP” (page 21) may suggest calcium metal NP (analogous to Ag NP, Cu NP etc., used in the paper reviewed).
Explanation: New abbreviation ‘HANPs’ added for hydroxyapatite nanoparticles.
- The reference list should be corrected. If the Authors have used DOI numbers, they should be added for all papers (when available). For many references, authors' names were given inconsistently with the MDPI style (e.g. [247], [249]‐[266]
Explanation: The reference has been revised. The DOI of the papers has been deleted.
- Please correct:
‐ “according to the estimation of Food and Agriculture Organization [2], the agriculture” (page 1);
‐ “nitrogen – sulphur – oxygen” (page 2);
‐ “MgO NPs” (page 3);
‐ “CuO NPs” (page 5);
‐ “Cu content” (page 5, 2x);
‐ “Helianthus annuus (sunflower)” (page 6);
‐ “siRNA” (page 6; it is an abbreviation from “small interfering RNA, not silicon RNA);
‐ “poly(aryl ethers)”, “poly(amidoamine)” (page 7; according to IUPAC recommendations, prefix “poly” should be used with brackets without space);
‐ “vitamin C” (page 8);
‐ “Xanthomonas axonopodis pv. vesicatoria” (page 13);
‐ “Furthermore, carbon nanotubes” (page 13);
‐ “Begomovirus” (page 13);
‐ "nanofertilizers, nanopesticides, nanofungicide, nanobioremediation and nanobased" (page 13);
‐ "1 μm", "150 nm" (pages 16 & 20; the International System of Units (SI) prescribes inserting a space between a number and a unit of measurement);
‐ “wheat”, “proline” (page 19);
‐ “ ‘Valencia’ “ (page 20; 2x)
‐ “and 120” (page 20);
‐ “lower Na+ and Cl‐ concentrations” (page 20);
‐ “proteins – heat shock proteins (HSP)” (page 21).
Explanation: The above suggestions have been incorporated into the revised manuscript.

Reviewer 3 Report
Ding et al. have revised the current stage of the nanotechnology applied to agriculture. The topic is quite interesting. However, the manuscript is not well organized and it could be difficult to follow by the reader. In addition, several examples of the effect of nanoparticles supplied to different crops to crop nutrition or protection, sensor, etc… have given along the manuscript. However, an explanation of the action mechanism, comparison with the behaviour of the bulk counterpart or with other nanoparticles are missing. This information is relevant for the reader. The following points must be corrected/changed:
p.2. “NPs is only released in the required quantity and reach specific cellular organelles” This sentence should be soft. In addition, the behaviour of the nanoparticles depends strongly on the chemical composition, particle size, function…
In the first section, authors explain some beneficial effects of each kind of nanoparticles (Zinc, copper, silver, etc…). In the next section, authors give examples of the beneficial effects of the same nanoparticles with specific applications. However, examples of these application had been explained in the first section. Later, in the section 4, authors talk about the beneficial effects in other fields. However, they do not give any example of some specific applications ( like in the case of nanofertilizers or nanopesticides). In this regard, authors should organized the manuscript on the basis of one specific topic (kind of nanoparticle or function). It would make easier to follow the manuscript.
p.9 “Chitosan NPs based release system has been employed for the control and targeted delivery of nitric oxide” Is nitric oxide a plant growth regulator?
p.9 “Micro-particulates” Nanoparticles are in the nanometric range, not in the micrometric range.
p. 9. “A crystalline porous polymer material was used for PGRs detection in…” Authors should indicate the material they mean.
p.10. Any example of specific nanoparticles is given by the author related to the role of nanotechonoly in plant genetic engineering.
p.11 “A DNA-nanostructure” What authors mean? Is it a nanoparticle conjugated to DNA? If yes, which kind of nanomaterial? (composition, size, etc…)
p.13. “ The agriculture control can be monitored by nanotechnoly, especially trough tis microsize” Nanotechnology is related to nanosize
p.15. Section 4.1. Authors do not give any example of nanomaterials as nanofertilizer
p.16. . Section 4.3. Authors do not give any example of nanomaterials as nanopesticide
p.16. “To overcome these limitations, nano-pesticides are the best viable alternative to conventional pesticides” This sentence is too strong. Authors do not give reasons to support this sentence.
P.16. “1 μm nanoparticle form” Nanoparticle is the nanometric range.
p.17. Section 4.4. No examples about nanosensors are given.
Author Response
Response to reviewer#3
Ding et al. have revised the current stage of the nanotechnology applied to agriculture. The topic is quite interesting. However, the manuscript is not well organized and it could be difficult to follow by the reader. In addition, several examples of the effect of nanoparticles supplied to different crops to crop nutrition or protection, sensor, etc… have given along the manuscript. However, an explanation of the action mechanism, comparison with the behaviour of the bulk counterpart or with other nanoparticles are missing. This information is relevant for the reader. The following points must be corrected/changed:
p.2. “NPs is only released in the required quantity and reach specific cellular organelles” This sentence should be soft. In addition, the behaviour of the nanoparticles depends strongly on the chemical composition, particle size, function…
Explanation: The sentence has been reframed.
In the first section, authors explain some beneficial effects of each kind of nanoparticles (Zinc, copper, silver, etc…). In the next section, authors give examples of the beneficial effects of the same nanoparticles with specific applications. However, examples of these application had been explained in the first section. Later, in the section 4, authors talk about the beneficial effects in other fields. However, they do not give any example of some specific applications ( like in the case of nanofertilizers or nanopesticides). In this regard, authors should organized the manuscript on the basis of one specific topic (kind of nanoparticle or function). It would make easier to follow the manuscript.
Explanation: We have deleted the overlapping portions from the section 4.0. Section 2.0 describe the type and function of various NPs. We have also deleted the overlapping portion from the section 1.0. Now the sections different information.
p.9 “Chitosan NPs based release system has been employed for the control and targeted delivery of nitric oxide” Is nitric oxide a plant growth regulator?
Explanation: No. Nitric oxide (NO) is a signalling molecule involved in plant response to various abiotic stresses.
p.9 “Micro-particulates” Nanoparticles are in the nanometric range, not in the micrometric range.
Explanation: The sentence has been corrected.
- 9. “A crystalline porous polymer material was used for PGRs detection in…” Authors should indicate the material they mean.
Explanation: [Fe3O4@COF(TpDA)] added.
p.10. Any example of specific nanoparticles is given by the author related to the role of nanotechonoly in plant genetic engineering.
Explanation: The example like carbon based nanoparticle (single and multiwalled carbon nanotubes), metal based NPs etc. along with other examples have been already discussed (page 11-12).
p.11 “A DNA-nanostructure” What authors mean? Is it a nanoparticle conjugated to DNA? If yes, which kind of nanomaterial? (composition, size, etc…)
Explanation: DNA nanostructure are nanoscale structure made of DNA, which acts both as a structural and functional element. DNA nanostructure can serve as scaffolds for the formation of more complex structures. It can internalize into plant cells and deliver siRNA to mature plant tissue without external aid.
p.13. “ The agriculture control can be monitored by nanotechnoly, especially trough tis microsize” Nanotechnology is related to nanosize.
Explanation: Corrected as “Nanosize”.
p.15. Section 4.1. Authors do not give any example of nanomaterials as nanofertilizer
Explanation: We have added the name of some approved nanofertilizer used in the world today for example; Nano Ultra-fertilizer, Nano capsule, Nano max NPK, TAG NANO etc. A complete list has been added in the section 4.1.
p.16. . Section 4.3. Authors do not give any example of nanomaterials as nanopesticide
Explanation: Example of nanopesticides such as Nnao green, encapsulated plant protection agent, Agro nanotechnology crop etc. has been added in the section.
p.16. “To overcome these limitations, nano-pesticides are the best viable alternative to conventional pesticides” This sentence is too strong. Authors do not give reasons to support this sentence.
Explanation: A paragraph has been added to support the claim.
P.16. “1 μm nanoparticle form” Nanoparticle is the nanometric range.
Explanation: “This approval covers synthetic amorphous silicon dioxide as a nanomaterial in the form of stable aggregated particles of particle size > 1µm, with primary particles of nanosize”.The sentence has been reframed for more clear meaning.
p.17. Section 4.4. No examples about nanosensors are given.
Explanation: A paragraph containing various nanosensor has been added in section 4.4..

Round 2
Reviewer 3 Report
Ding et al. have introduced some modifications, in comparison with the previous version of the manuscript, to answer the different points of the previous review I provided. Although an improvement has been appreciated, some points have not been properly addressed or discussed.
1. As I exposed in the last revision, an explanation of the action mechanism, comparison with the behaviour of the bulk counterpart or with other nanoparticles are missing along the manuscript, making difficult to identify the real advantages of the nanoparticles that authors have included in the manuscript.
2. Authors have maintained the structure of the manuscript. As I exposed in the last review, some sections give information about the same topic but in different part of the manuscript. This fact makes difficult to read the manuscript. In the section 2.1, authors discussed partially the specific function of some nanoparticles (i.e ZnO, TiO2, etc…). Other examples of the effects of these nanoparticles in specific roles in agriculture are found in subsequent sections of the manuscript. It would be easier for readers if the information is organized according to the kind of nanoparticles, or according to the specific function of the nanoparticles.
3. In the previous state of the manuscript, some sections did not show any example about the role of nanoparticles on a specific function (i.e. nanofertilizers, nanopesticides…). In the new version, authors have included some commercial materials, labelled as nanoparticles, with specific role in these functions. However, any information about these nanomaterials (particle size, morphology, etc…) or effects on crops are given. Indeed, some of these examples are not accompanied by a bibliographic reference, making difficult to look for the information of these materials in the literature.
Apart from the information exposed above, there are some minor points I found in the new version of the article:
1. In the last revision, I ask author if nitric oxide was a plant growth regulator. Their answer was no. In this regard, why the include an example of nitric oxide releasing nanomaterials in the section 3.2 Role of nanotechnology for delivery of plant growth regulators (PGRs).
2. In the same page, they include an example (ref. 128) about microsphere of chitosan. If the material is a microsphere, it is not a nanoparticle. Therefore, this example must not be included in the review.
On the basis of the information exposed above, I consider that the manuscript is the current state is not adequate for the publication in nanomaterial.
Author Response
Respond to the reviewer
Ding et al. have introduced some modifications, in comparison with the previous version of the manuscript, to answer the different points of the previous review I provided. Although an improvement has been appreciated, some points have not been properly addressed or discussed.
- As I exposed in the last revision, an explanation of the action mechanism, and comparison with the behavior of the bulk counterpart or with other nanoparticles are missing along the manuscript, making difficult to identify the real advantages of the nanoparticles that authors have included in the manuscript.
Explanation: A new paragraph (1 page) (section 6.0) has been added to the revised manuscript. Exploring the plant-nanoparticle interactions. The new section discusses the mode of uptake of NPs, their translocation and accumulation. A brief discussion on the movement of NPs within the plant has also been discussed.
- Authors have maintained the structure of the manuscript. As I exposed in the last review, some sections give information about the same topic but in different part of the manuscript. This fact makes difficult to read the manuscript. In the section 2.1, authors discussed partially the specific function of some nanoparticles (i.e ZnO, TiO2, etc…). Other examples of the effects of these nanoparticles in specific roles in agriculture are found in subsequent sections of the manuscript. It would be easier for readers if the information is organized according to the kind of nanoparticles, or according to the specific function of the nanoparticles.
Explanation: Considering the above suggestions the following changes have been carried out.
- Section 2 has been shortened as only a brief introduction to NPS. The types of NPs and synthesis of NPs have been presented through the diagram.
- The discussion about each type of NPs and its functional role has been removed from this section to avoid confusion.
- Section 3 has been organized on the specific role of nanotechnology in agriculture. This section also deals with the role of different NPs in agriculture, including the required information from old section 2.
- Section 4: The aim of this section is to present the role of tailored nanomaterials in agriculture. The available or commercial nanomaterials for example have also been discussed in this section. The information presented here is not overlapping with sections 2&3.
- We hope the above arrangements of the section will be helpful to the readers.
- In the previous state of the manuscript, some sections did not show any example of the role of nanoparticles on a specific function (i.e. nanofertilizers, nanopesticides…). In the new version, the authors have included some commercial materials, labeled as nanoparticles, with a specific role in these functions. However, any information about these nanomaterials (particle size, morphology, etc…) or effects on crops are given. Indeed, some of these examples are not accompanied by a bibliographic reference, making it difficult to look for the information of these materials in the literature.
Explanation:
- The reference for these commercially available products (i.e. Nanofertilizers, nanopesticides, nanofungicides, nanobiosensors etc.) are has been added to the revised manuscript.
- The specific role of tailored nanomaterials is discussed in Section 4.0.
- Little is known about these commercially available products regarding their particle size, morphology, etc., or their effects on the crop. And therefore it isn't easy to provide data specifically on these in the form of literature on commercially available products. More research needs to be done.
Apart from the information exposed above, there are some minor points I found in the new version of the article:
- In the last revision, I ask author if nitric oxide was a plant growth regulator. Their answer was no. In this regard, why the include an example of nitric oxide releasing nanomaterials in the section 3.2 Role of nanotechnology for delivery of plant growth regulators (PGRs).
Explanation: The line has been deleted in the revised manuscript.
- In the same page, they include an example (ref. 128) about microsphere of chitosan. If the material is a microsphere, it is not a nanoparticle. Therefore, this example must not be included in the review.
Explanation: The line has been deleted in the revised manuscript.

Round 3
Reviewer 3 Report
Authors have improved siginficantly the last versions of the manuscript. Consequently, i think that the current manuscript is appropiate for being published in Nanomaterials.